# Monolithic Integrated High Frequency GaN DC-DC Buck Converters with High Power Density Controlled by Current Mode Logic Level Signal

**Longkun Lai** [1,2], **Ronghua Zhang** [1,2], **Kui Cheng** [1,2], **Zhiying Xia** [1,2], **Chun Wei** [1,2], **Ke Wei** [1,2], **Weijun Luo** [1,2,*] **and Xinyu Liu** [1,2,*]

1   Institute of Microelectronics, Chinese Academy of Sciences, Beijing 100029, China; lailongkun@ime.ac.cn (L.L.); zhangronghua@ime.ac.cn (R.Z.); chengkui@ime.ac.cn (K.C.); xiazhiying@ime.ac.cn (Z.X.); weichun@ime.ac.cn (C.W.); weike@ime.ac.cn (K.W.)
2   School of Microelectronics, University of Chinese Academy of Sciences, Beijing 100049, China
*   Correspondence: luoweijun@ime.ac.cn (W.L.); xyliu@ime.ac.cn (X.L.)

**Abstract:** Integration is a key way to improve the switching frequency and power density for a DC-DC converter. A monolithic integrated GaN based DC-DC buck converter is realized by using a gate driver and a half-bridge power stage. The gate driver is composed of three stages (amplitude amplifier stage, level shifting stage and resistive-load amplifier stage) to amplify and modulate the driver control signal, i.e., CML (current mode logic) level of which the swing is from 1.1 to 1.8 V meaning that there is no need for an additional buffer or preamplifier for the control signal. The gate driver can provide sufficient driving capability for the power stage and improve the power density efficiently. The proposed GaN based DC-DC buck converter is implemented in the 0.25 μm depletion mode GaN-on-SiC process with a chip area of 1.7 mm × 1.3 mm, which is capable of operating at high switching frequency up to 200 MHz and possesses high power density up to 1 W/mm$^2$ at 15 V output voltage. To the authors' knowledge, this is the highest power density for GaN based DC-DC converter at the hundreds of megahertz range.

**Keywords:** bootstrapped capacitor; DC-DC buck converter; depletion-mode GaN HEMT; MMIC; power density

## 1. Introduction

The increasing requirements of power consumption, high power density and high operational frequency of modern applications have been appealing for converters with much smaller size and higher switching frequency. The demanding for a reduced converter volume is stimulated, especially, by the information technology applications where the rapid development of integrated circuit technology had aroused more compact systems with higher power dissipation [1]. A small volume means high power density which is equivalent to greater design freedom, lower installation cost and more system robustness. Traditional DC-DC converters are mostly implemented in the process of Si MOSFET (Metal-Oxide-Semiconductor Field-Effect Transistor) with extremely low $R_{ON}$ (on-state resistance) and high efficiency performance over hundreds of kilohertz [2–7] However, they can not operate at very high switching frequency with desirable power density due to the large parasitic capacitors [8]. Under such circumstances lots of efforts have been put in GaN based switching DC-DC converters [9,10] for higher operating frequency, breakdown voltage and power density performance comparing to the counterparts of Si devices [11,12].

For the GaN based switching DC-DC converter, depletion-mode devices usually have lower ON-resistance and smaller parasitic capacitance [13], which is more suitable for high frequency and

high power density demanding converters. To further use those merits, it is necessary to realize full integration of the DC-DC converter containing the power switching stage and gate driver. Then the size of the converter chips as well as the parasitic capacitance coming from the devices or introduced by the external package can be dramatically decreased. With discrete devices, Miguel R et al. [9] illustrated a high efficiency demonstrator which can operate at 10–40 MHz switching frequency, Nicolas et al., reported a converter operating at 50 MHz with a preamplifier to amplify the control signal (after amplifying, the swing is 6.27 V) [14] and for a higher frequency range Ming-Jie et al. demonstrated a converter that can operate at 300 MHz with an additional buffer for a control clock signal [15] but with low power density, i.e., $4.16 \times 10^{-6}$ W/mm$^2$. To realize the monolithic integrated GaN based DC-DC converter and improve operating frequency, Zhang et al. [10] used three circuit topologies at 100 MHz with a level shifter matching network to transmit the control signal (the swing is 5 V) generated by FPGA (Field Programmable Gate Array). Pilsoon et al. [16] reported a converter that can work at high switching frequency, i.e., 680 MHz with no need for a gate driver, but only with 0.24 W/mm$^2$ power density using the 0.25 μm GaN-on-SiC process. As enumerated, though the converters that can operate at hundreds of megahertz range have been realized no matter in integrated or discrete form, the gate control signal in some of them still need to be modified by additional buffer or a preamplifier and the power density is still very low at such a frequency range. This paper concentrates on the highly integrated GaN based DC-DC converter which can be controlled directly by the current mode logic (CML) level signal of which the swing is 0.7 V (from 1.1 to 1.8 V), operating at high switching frequency and possessing high power density. Two converter topologies with and without a bootstrapped capacitor structure are designed and analyzed. The driver integrated in the converter can amplify the CML level control signal (swing is 0.7 V) to a driving signal (swing is close to 25 V). The demonstrated GaN based DC-DC converter with a bootstrapped capacitor structure possesses higher performance comparing to the converter without bootstrapped capacitor in terms of efficiency and both of them exhibit 15 V of output voltage, 2.2 W of output power and 1 W/mm$^2$ of power density working at 200 MHz switching frequency. It is the first time that a GaN based DC-DC converter exhibits 1 W/mm$^2$ at hundreds of megahertz with the CML level control signal to the authors' knowledge.

This paper is organized as follows: Section 2 shows the working principal and simulated results of the driver and the integrated converters. Section 3 presents the experimental results of both converters. Conclusion and discussion are given in Section 4.

## 2. GaN Based Switching DC-DC Converter

As shown in Figure 1, the monolithic integrated GaN based switching DC-DC converter is composed of three parts: half-bridge switching power stage (transistors $T_{HS}$ and $T_{LS}$), gate drivers (the triangle parts) and inductive filter network ($L_R$ and $C_R$). The driver for $T_{HS}$ is a high side driver and the other one is a low side driver. In such a structure implemented with D-mode (normally on) GaN HEMT (High Electron Mobility Transistor), the design of the high side driver is quite a challenging issue. It is because the high side transistor $T_{HS}$ need a high driving voltage swing from −5 V (to turn it off) to the required output voltage (to turn it on) whereas a comparatively small voltage swing from −5 to 0 V is needed for the low side transistor $T_{LS}$. This section demonstrated two approaches to achieve the difficult target with small swing control signal at a very high frequency and high power density.

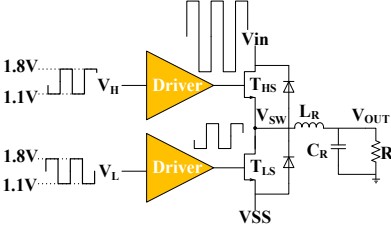

**Figure 1.** GaN switching DC-DC topology.

### 2.1. Driver Design

The schematic of the driver is illustrated in Figure 2 consisting of the first amplitude amplifier stage, the second level shifting stage and the third resistive-load amplifier stage. It can provide a suitable driving signal for $T_{HS}$ (close to 25 V swing) and $T_{LS}$ to turn them on and off by amplifying and modulating the very small CML level control signal (0.7 V swing), which can solve the challenging issue of the switching DC-DC converter. The 1st and 2nd stage of both side driver have the completely identical topology, devices (no matter passive or active components) and bias voltage (VDD× or VSS×) as shown in Figure 2.

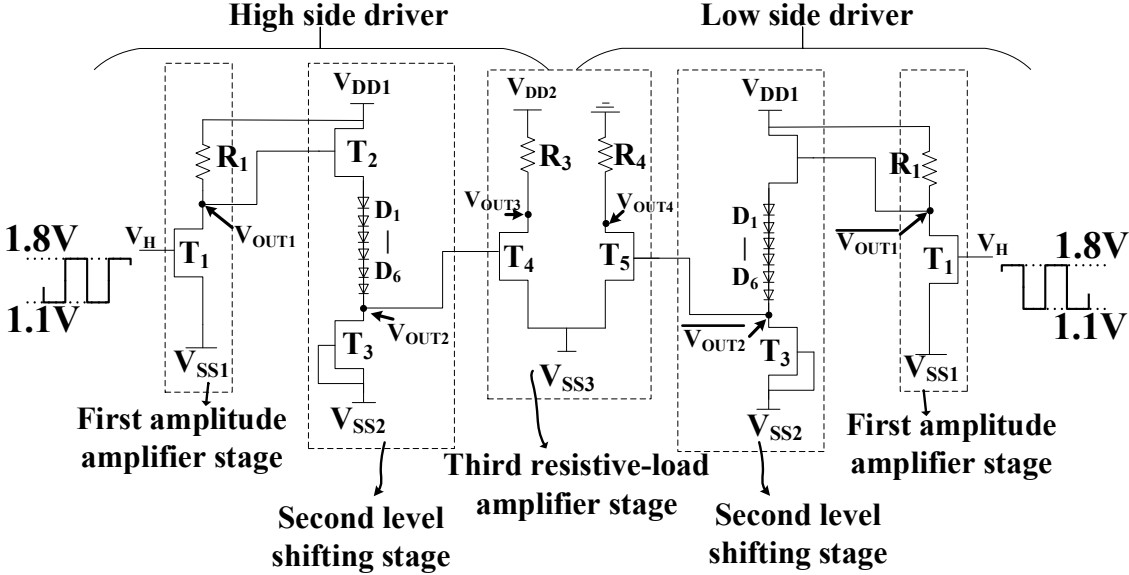

**Figure 2.** Schematic of the driver including a high side driver ($V_{OUT3}$ to drive $T_{HS}$) and low side driver ($V_{OUT4}$ to drive $T_{LS}$).

For the 1st stage, its function is to amplify the driver's CML level control signal (of which the amplitude is from 1.1 to 1.8 V) to a bigger swing (5 V). Additionally, for the 2nd stage, it will shift the pulse from the output of the 1st stage to a suitable level, which can offer enough driving ability to drive the third stage to be the ON state and OFF state. Additionally the third part of the driver is a resistive-load amplifier stage to amplify the signal from the output of the 2nd stage to drive the power stage. Compared to discrete converters which use bias tree to enlarge a driving signal in [14] and a hybrid gate driver in CMOS (Complementary Metal Oxide Semiconductor) [15], and integrated converters [10] with an external level shifter matching network, the proposed driver in this paper can be controlled directly by the CML level signal. The driver has less demand for a drivers' control signal and the 1st and 2nd stage can offer over a 4 V swing for the input of the 3rd stage. Subsequently $T_4$ and $T_5$ in the 3rd stage will turn on and off more thoroughly, which means the driver owns much more powerful driving capability. In such situation, the power stage will possess better time domain performance and higher power density. To the authors' knowledge, it is the first time to report the proposed three stages driver structure in the DC-DC converter.

A simplified diagram for the operation of the high side driver is shown in Figure 3 and the low side driver has the same structure and operation mode except for the inverted input signal so it has been hided due to the area limit. When $V_H$ is high(1.8 V), $T_1$ is ON of which the ON-state resistor is $R_{T1, on}$ and $V_{OUT1}$(output of the 1st stage for the high side driver) is close to $V_{SS1}$ as shown in Figure 3a and $\overline{V_{OUT1}}$ (output of the 1st stage for the low side driver) is close to VDD1. For the second level shifting stage, transistor $T_3$ is equivalent to a current source($I_{T3}$) for the connection of its gate and

source which means the $V_{GS}$ of $T_3$ is zero so that $T_3$ will maintain the ON state and the bias current can be figured out by

$$I \approx \frac{1}{2}\frac{W}{L}\mu_n C_{ox}(0 - V_{TH})^2 \tag{1}$$

where $W$, $L$, $\mu_n$ and $C_{ox}$ are the technology parameters. For the Schottky diodes $D_1$-$D_6$, each of them have the same voltage decrement, set to $V_D$ which is 2 V at 20 mA bias current, the total voltage decrement of the six diodes is $6 \times VD$. The magnitude of the voltage decrement can be controlled by adjusting the diameter of the diodes to get the proper $V_{OUT2}$ (output of the 2nd stage) by

$$V_{OUT2} = \left(V_{OUT1} - V_{GS,T_2}\right) - 6 \times V_D \tag{2}$$

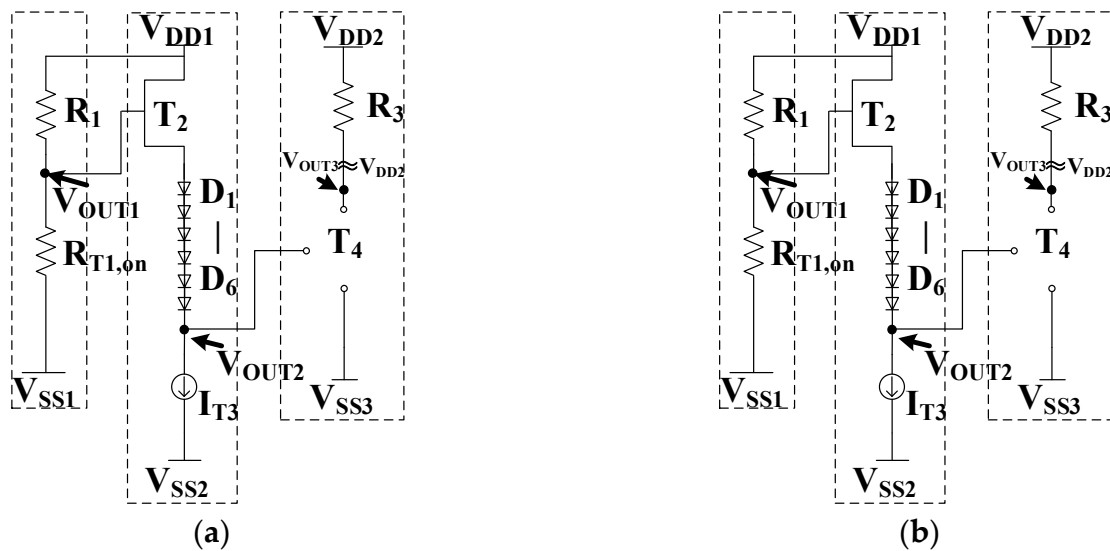

**Figure 3.** Simplified diagram for the operation of the high side driver, (**a**) $V_{OUT3}$ output high level (close to VDD2) and (**b**) $V_{OUT3}$ output low level (close to VDD2).

Additionally then, comparatively low $V_{OUT}$ of the 2nd stage will be transmitted to the left side of the third stage and control $T_4$ to turn off, which make the $V_{OUT3}$ (output of left side for the 3rd stage) output a high level voltage nearly approaching $V_{DD2}$ (bias voltage of the left side for the 3rd stage). Simultaneously the low side driver accepts a low level signal (1.1 V) and outputs a low level voltage which is $V_{OUT4}$ (output of right side for the 3rd stage) closing to $V_{SS3}$ (common bias voltage of the 3rd stage). Vice versa, when $V_H$ is low (1.1 V) and $V_L$ is high (1.8 V) as shown in Figure 3b, $V_{OUT3}$ will be close to $V_{SS3}$ and $V_{OUT4}$ will be close to GND.

The rise and fall time of the output waveform are optimized through adjusting the resistors of each stage. So for the 1st stage, the load resistor $R_1$ will determine the rise time of $V_{OUT1}$ (output of the 1st stage) thereby affecting the counterpart of $V_{OUT}$ for the converter. Figure 4a shows the simulated waveform of $V_{OUT1}$ under conditions of different $R_1$. Making a compromise between the frequency performance and power dissipation, we chose $R_1$ equaling to 500 Ω. As for load resistors in the third differential stage, with the same standpoint like the 1st stage, we chose 500 Ω for $R_3$ in the left side and 250 Ω for $R_4$ in the other side. The simulated output waveforms of both side with different resistors are shown in Figure 4b,c. After confirming the value of all the resistors, the simulated optimized results for driver are shown in Figure 5 proving that the demonstrated gate driver can satisfy the demand for a high swing of the high side power transistor $T_{HS}$ at 200 MHz, not to mention $T_{LS}$.

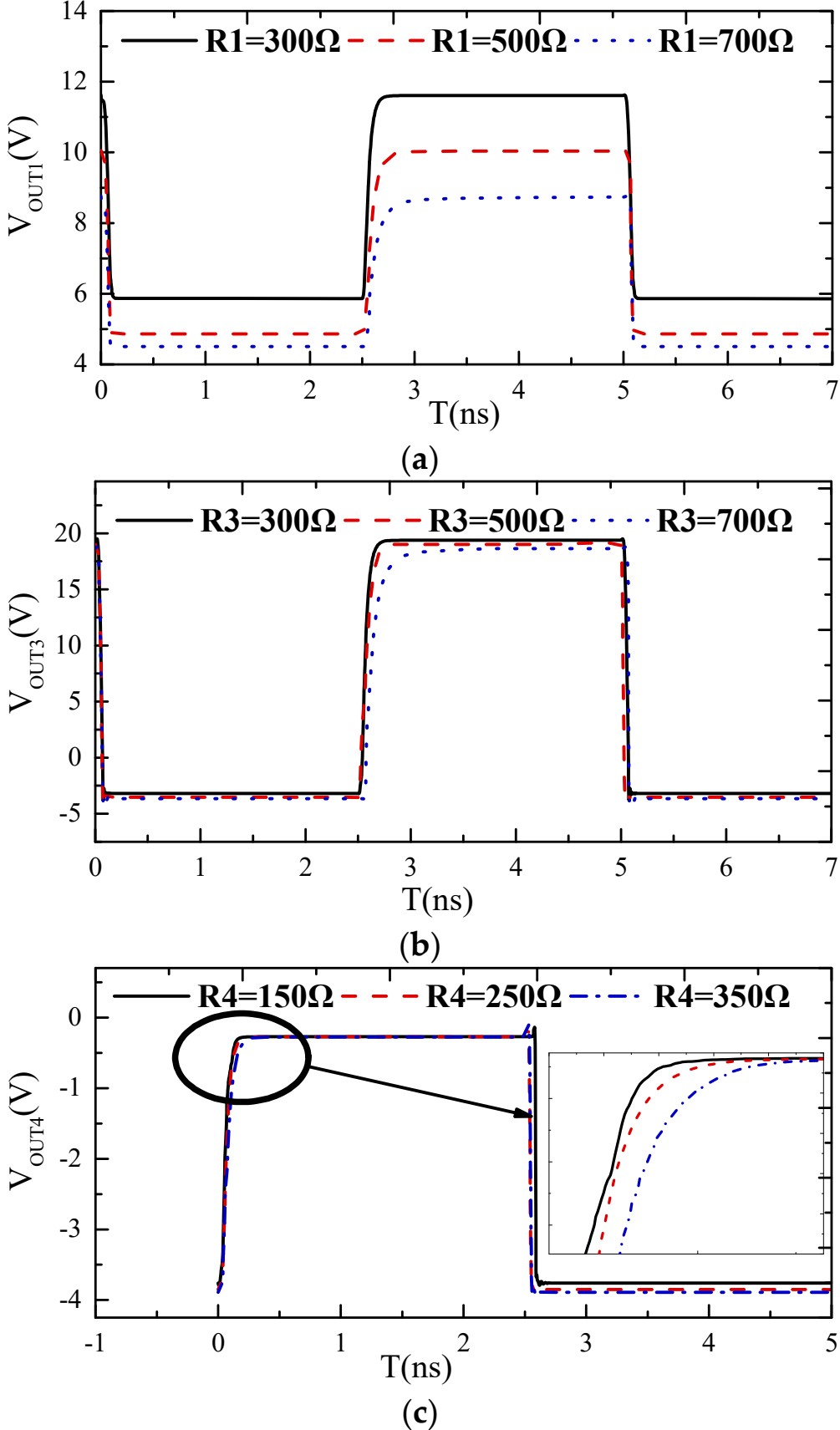

**Figure 4.** The simulated output waveform: (**a**) 1st stage, (**b**) left side of the 3rd stage and (**c**) right side of the 3rd stage.

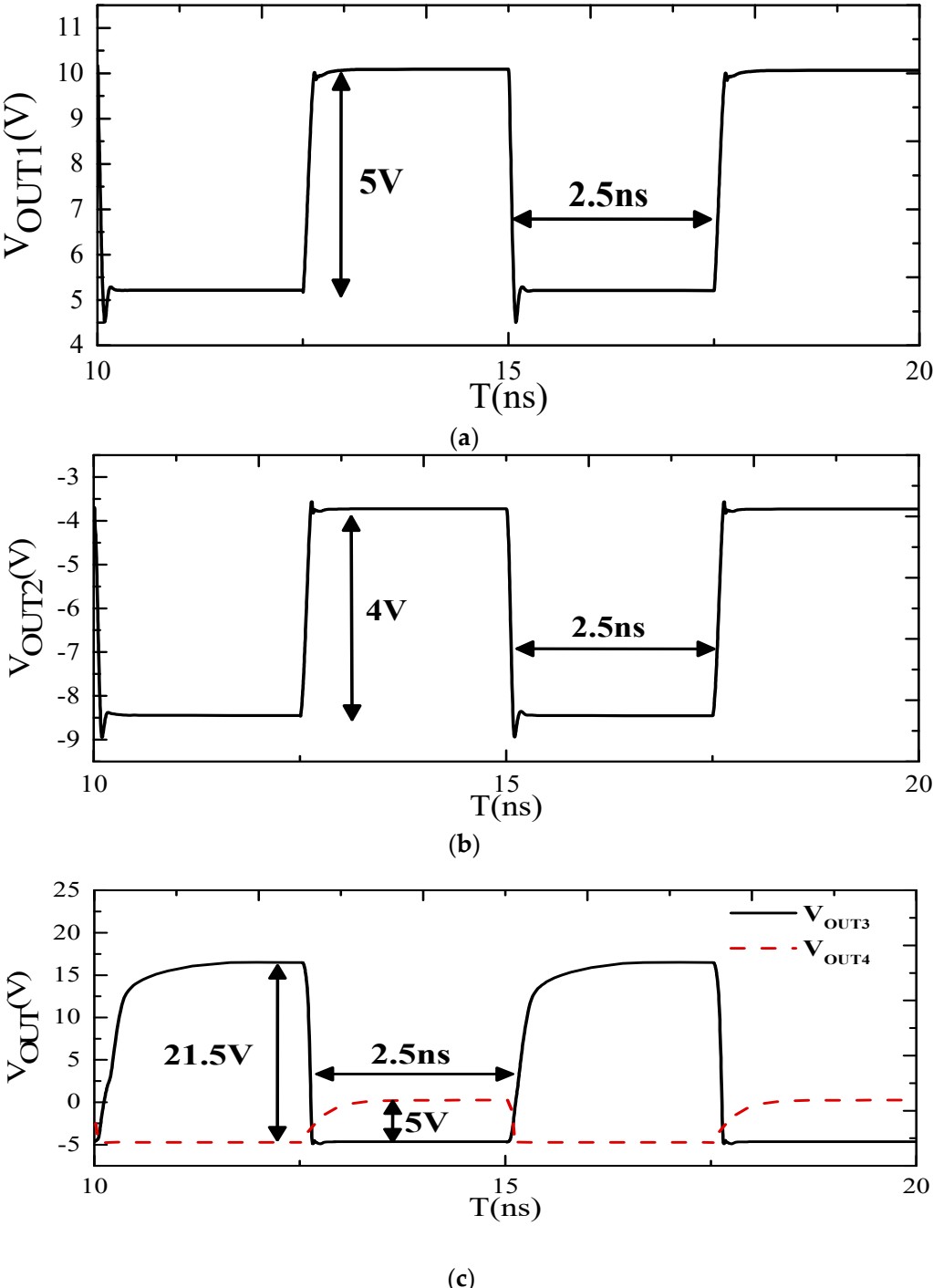

**Figure 5.** The simulated results of gate driver: (**a**) $V_{OUT1}$ ($V_{OUT}$ of the 1st stage), (**b**) $V_{OUT2}$ ($V_{OUT}$ of the 2nd stage) and (**c**) $V_{OUT3}$ and $V_{OUT4}$ ($V_{OUT}$ for the left and right side of the 3rd stage respectively).

### 2.2. Converter Design

After verifying the function and driving capability of the driver, the power stage was integrated with the driver in one chip to decrease the parasitic parameters. Figure 6 shows the schematic of the monolithic integrated GaN based DC-DC converter. Due to the area limit, this schematic just draws the high side driver and simplifies the 1st and 2nd stage of the low side driver with a rectangle. The half-bridge power stage was composed of the high side transistor $T_{HS}$ and the low side transistor $T_{LS}$. The gate of $T_{HS}$ and $T_{LS}$ was connected to $V_{OUT3}$ (output for the left side of the 3rd stage) and

$V_{OUT4}$ (output for the right side of the 3rd stage) respectively. When $T_{HS}$ once receives a high level of gate voltage, which comes from $V_{OUT3}$ closing to Vin (18 V), it will be the ON state, meanwhile the $T_{LS}$ receives a low gate voltage and turns off, eventually the output level will be high, at 15 V in this paper. To maintain the floating gate voltage of $T_{HS}$ when it is ON, $V_{DD2}$ (18.5 V) was set to be slightly bigger than Vin (18 V). Vice versa, when $V_{OUT3}$ is low and $V_{OUT4}$ is high, the output voltage of the total circuit will be low to 0 V because the source of $T_{LS}$ is GND. Figure 7 shows the simulated waveform of $V_{SW}$ (black solid line). The result proved that the designed converter could transfer 18 V to 15 V with the CML level control signal at 200 MHz. However, the simulated $V_{DD2}$ was near or bigger than Vin and the simulated power dissipation for the left side of the 3rd stage was 0.53 W resulting in the deterioration of the overall efficiency.

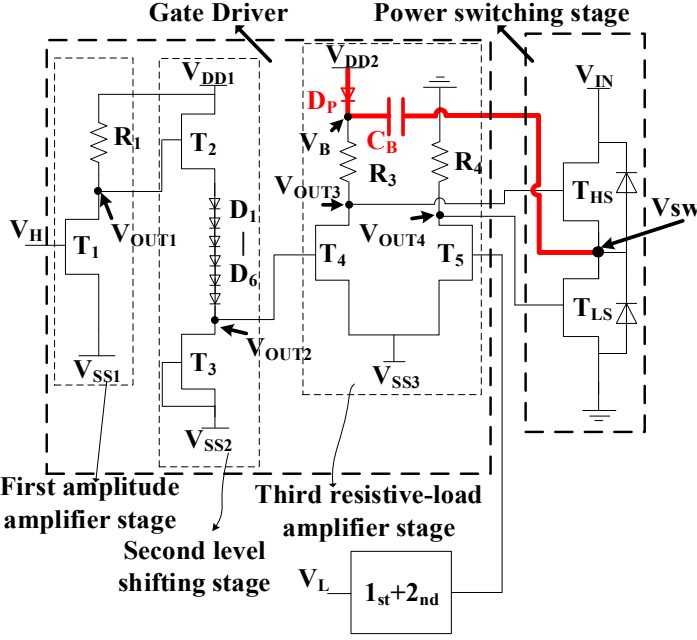

**Figure 6.** Schematic of two topologies for GaN DC-DC converter: without the bootstrapped capacitor (not including the red bold line path) and with the bootstrapped capacitor (include the red bold line path).

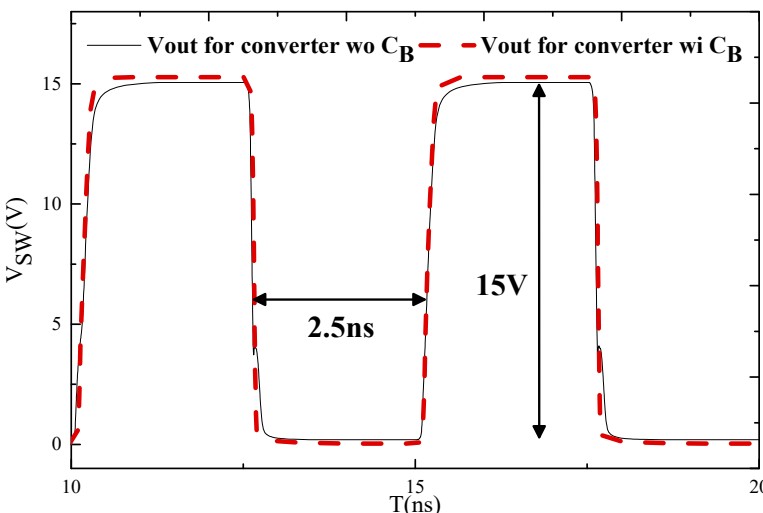

**Figure 7.** The simulated results for $V_{SW}$ of both converters: black solid line for the converter without $C_B$ and red dash line for the converter with $C_B$.

So to reduce this part DC power dissipation, this paper proposed another topology with a bootstrapped capacitor as shown in Figure 6 (include the red bold line path). The red components (protect Schottky diode $D_P$ and bootstrapped capacitor $C_B$) and bold line form the so called bootstrapped path. The only difference in topology of the two types of converters is whether they have bootstrapped capacitor $C_B$ and the protecting diode Dp (the red bold line path) or not. When $T_{LS}$ is turned on, $C_B$ will be charged to $V_{DD2}$ (which is 3 V in the simulated result of the bootstrap topology) and the output voltage level will be nearly zero. Then after $T_{LS}$ was turned off, $V_{OUT}$ of the converter increased so $V_B$ (the upper plate potential of $C_B$) increased too, to maintain the charge between the two plates of $C_B$. Additionally different results of the output for the third stage ($V_{OUT3}$) were given with different $C_B$ values, this paper eventually chose 100 pf for a better time domain and bootstrapped performance as depicted in Figure 8. The $V_{DD2}$ of the converter with a bootstrapped capacitor ($C_B$) was much smaller (3 V) than the counterpart in the converter without a bootstrapped capacitor (18.5 V) in the simulated result. The simulated power dissipation for the left side of the 3rd stage was 0.12 W, which was much smaller than the counterpart in the converter without $C_B$ (0.53 W). With small $V_{DD2}$, the output of the left side for 3rd stage could still satisfy the demanding swing for $T_{HS}$. The simulated output waveform of the switching node ($V_{SW}$) for the converter with bootstrapped capacitor is shown in Figure 7 at 200 MHz. Meanwhile in Figure 7 the output high level of the converter with $C_B$ was slightly bigger than the converter without $C_B$. That is because the driver of the converter with $C_B$ had a much more powerful driving capability. Both of them had over 2.2 W output power at 200 MHz in the simulated results and can be driven simply by the CML level control signal.

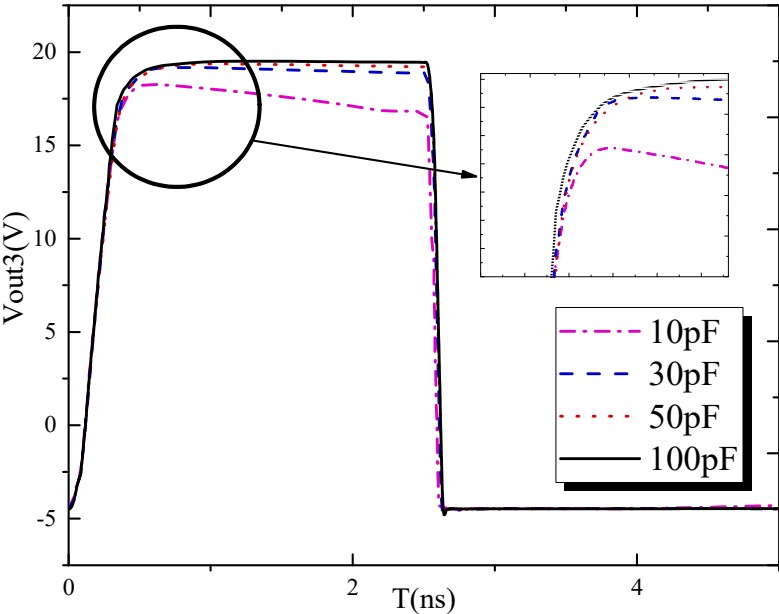

**Figure 8.** $V_{OUT3}$ with different $C_B$ values.

## 3. Experiment Results

The designed GaN based DC-DC converter is realized in the process of 0.25 μm GaN-on-SiC and the detailed description of the process can refer to [17]. The epitaxial structure of the AlGaN/GaN HEMT consists of a 3in SiC substrate, a 2 μm GaN buffer, a 1nm AlN interlayer, a 23 nm $Al_{0.23}Ga_{0.77}N$ barrier and a 2-nm GaN cap. The gate width of $T_1$, $T_4$, $T_5$ and $T_2$, $T_3$ was 2 × 125 μm and 20 μm respectively. The radius of the Schottky diodes used for level shifting was 20 μm. For the power stage, the gate width of $T_{HS}$ and $T_{LS}$ was both 8 × 125 μm to meet the demanding for a low output capacitor and low $R_{ON}$. The test of the converters was carried out by mounting the chips on PCB, which were connected through bonding wires. The CML level control signal (1.1–1.8 V) was generated

by an Agilent 81250 Parallel Bit Error Ratio Tester(PBER) with a sampling rate of 10.6 Gbps. The DC power supply was supported by HP4142B and HP6654A and the output waveform was measured by a 50 Ω-high-speed Lecroy SDA 816Zi-A oscilloscope with a 40GS/s sampling rate in the time domain.

First, the separated gate driver was tested, and its output waveform of the left side for the 3rd stage is shown in Figure 9. The gate driver could offer a pulse that the voltage swing could be up to 24.1 V (from −6.4 to 17.7 V) under 1.25 MHz with 150 ns for rising time and 30.7 ns for average falling time. The rising time was determined by $R_3 \times C_{OUT3}$ and the falling time was determined by $R_{ON} \times C_{OUT3}$, where $R_3$ is the left side load resistor of the 3rd stage, $R_{ON}$ is the ON-state resistor of left transistor of the 3rd stage and $C_{OUT3}$ is the equivalent output capacitor of the output node for the left side of the 3rd stage. Due to the large time constant, the separated driver can not be tested at higher frequency. However, the monolithic integrated converter could operate at 200 MHz resulting from the equivalent output resistance($R_E$) in the $V_{OUT3}$ node decreasing. Since when the driver is tested separately, its load resistor was 1 MΩ to simulate the gate of $T_{HS}$ leading to a large time constant ($R_3 \times C_{OUT}3$, $R_E$ was equal to $R_3$). While for the integrated circuit, its load resistor was 50 Ω, which means the time constant was equal to $R_E \times C_{OUT3}$, which was much smaller than $R_3 \times C_{OUT3}$ ($R_E$ was the parallel of $R_3$ and oscilloscope resistor (50 Ω) here).

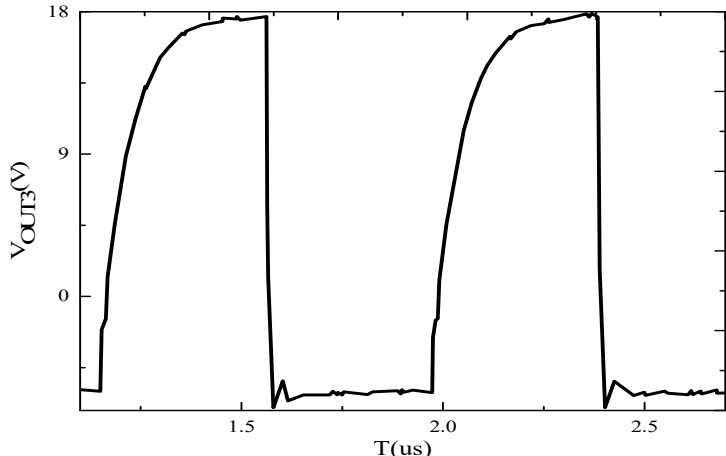

**Figure 9.** The experimental output ($V_{OUT3}$) for the left side of the gate driver where the bias voltage ($V_{DD2}$) of the left side for the 3rd stage is 18.5 V.

Figure 10 shows the prototypes for both converters. Additionally the comparison between the experimental output waveforms of two converters and the simulated output waveform of the converter with $C_B$ is given in Figure 11 at 200 MHz. Both of the two converters exhibited the highest experimental output level of 15 V. Comparing with the counterparts in [10] at 100 MHz, the waveform result of this paper was much smoother with less ripple at a higher frequency of 200 MHz. There is a phenomenon that should be explained that the edge of simulated and experimental waveforms is not coincident very well. Such a deviation is because the duty of the control signal generated by PBER was not literally 50% result from the coding mode of PBER. However, their qualitative behaviors are the same as shown in the black short-dash rectangle of Figure 11.

In Figure 12, the performance of efficiency and output power for two types of switching converters is depicted. The highest output power for the converter with a bootstrapped capacitor was 2.22 W, consequently its power density was 1 W/mm$^2$ and efficiency was 54.8% when Vin was 18 V. Apparently both converters had similar output power whereas the converter with $C_B$ had higher PAE (overall efficiency or power added efficiency) and DE (drain efficiency, i.e., power stage efficiency). It is because the use of the bootstrapped capacitors could dramatically decrease the demand for high $V_{DD2}$ (from 18.5 V in a converter without $C_B$ to 3 V in a converter with $C_B$) to reduce the power consumption of the driver and increase PAE over 5%.Therefore, the converter with $C_B$, which this paper proposed, had the

merits in requiring lower driver power dissipation and possessing higher DE and PAE, as expected, than the counterparts of the converter without $C_B$ under the approximate output power and power density condition.

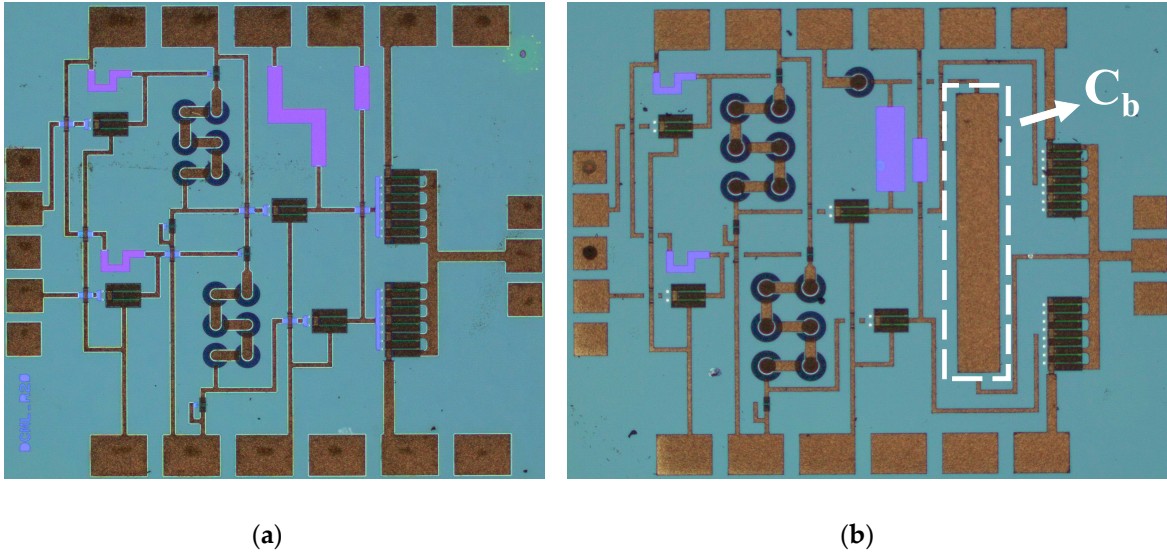

(**a**)　　　　　　　　　　　　　　　　　　　　　　　　(**b**)

**Figure 10.** Photos of the convertor chip prototype of: (**a**) convertor without a bootstrapped capacitor, 1.7 mm × 1.3 mm, and (**b**) convertor with a bootstrapped capacitor, 1.7 mm × 1.3 mm.

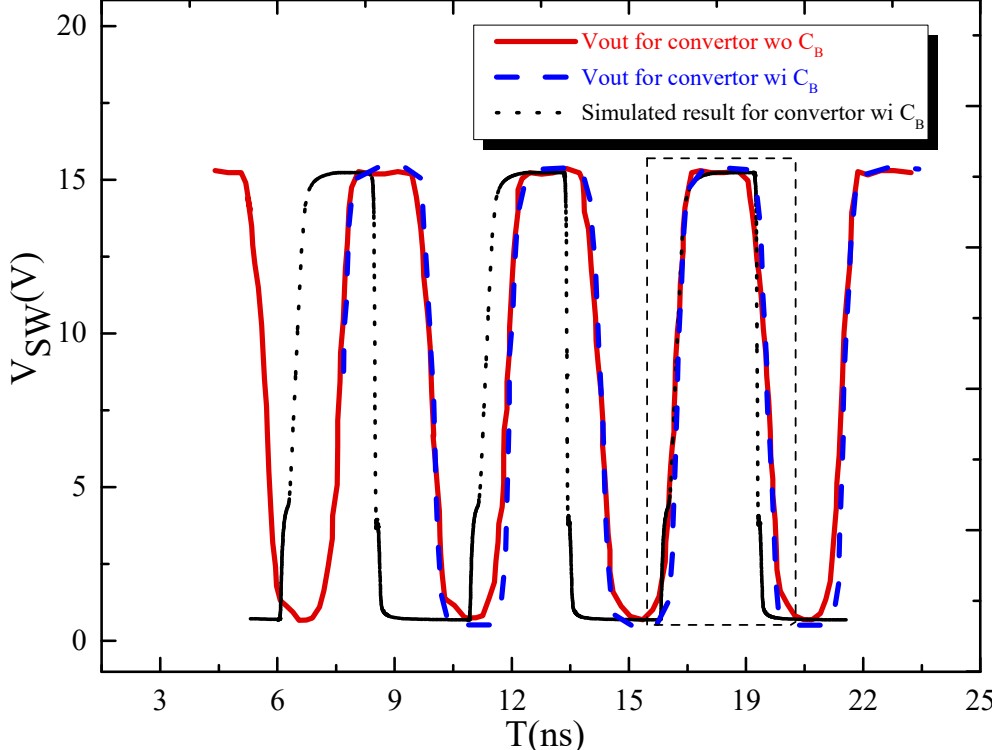

**Figure 11.** Comparison between the experimental output waveforms (of a converter with a bootstrapped capacitor and converter without a bootstrapped capacitor) and simulated output waveform (of a converter with a bootstrapped capacitor).

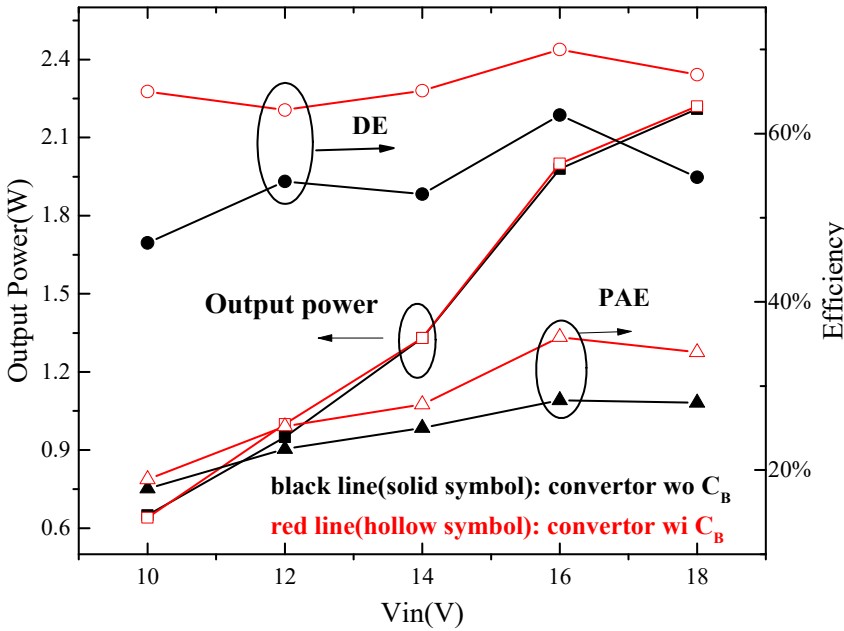

**Figure 12.** Measured efficiency and output power with differential.

Figure 13 demonstrates the power density performance comparison between the designed converter with a bootstrapped capacitor and the state-of-the-art buck converter. It was proven the proposed topology with a new driver and bootstrapped capacitor possessed the highest power density at a hundreds of megahertz range to the authors' knowledge. Table 1 shows the summary of the proposed circuit performance and a comparison with the previous DC-DC converters' results, illustrating that the designed converter of this work could be directly driven by a small control signal (CML: swing was 0.7 V) at a hundreds of megahertz range with the highest power density.

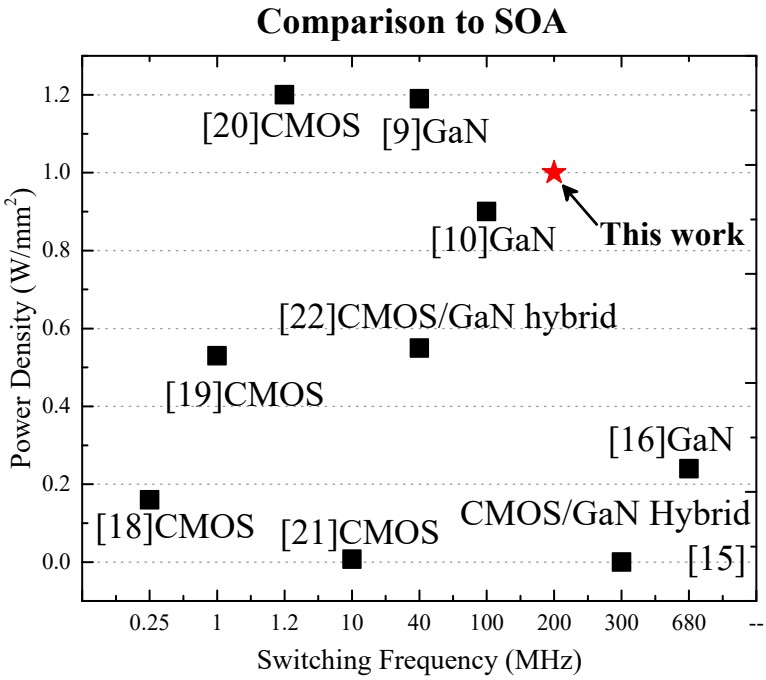

**Figure 13.** Comparison of this work to the state-of-the-art [9,10,15,16], [18–22] pulse signals at 200 MHz.

**Table 1.** Performance comparison with previous DC-DC converters.

| Symbol | [10] | [14] | [15] | This Work |
|---|---|---|---|---|
| Technology | 0.15 um GaN | Discrete GaN | 0.25 um GaN | 0.25 um GaN |
| Gate driver | integrated | N/A | CMOS driver | integrated |
| Control signal swing | 5 V | 6.27 V | N/A | 0.7 V |
| Frequency | 100 MHz | 50 MHz | 300 MHz | 200 MHz |
| Max converter efficiency | 88% | 90% | 47.3% | 54.8% |
| Area | $2.4 \times 2.3$ mm$^2$ | N/A | $0.94 \times 0.98$ cm$^2$ | $1.7 \times 1.3$ mm$^2$ |
| Power Density | 0.9 W/mm$^2$ | N/A | $4.4 \times 10^{-6}$ W/mm$^2$ | 1 W/mm$^2$ |

## 4. Conclusions

This paper demonstrated a monolithic integrated GaN based DC-DC buck converter, which can be directly controlled by the CML level signal of which the amplitude was from 1.1 to 1.8 V, with bootstrapped topology transferring 18 V to 15 V. The size of the chip was 1.7 mm $\times$ 1.3 mm of which the power density was 1 W/mm$^2$ and output power was 2.2 W with 54.8% power stage efficiency operating at 200 MHz. To the authors' knowledge, this was the highest power density for a GaN based DC-DC converter at a hundreds of megahertz range by using the CML level control signal.

**Author Contributions:** L.L. completed the circuit design and test, obtained and analyzed the measurement data, and wrote the manuscript. R.Z., Z.X., C.W. and K.C. assisted in the implementation and simulation for the circuits. W.L. proposed the schematic concept and was in charge of review and editing. W.L., X.L. and K.W. was the project administrators. All authors have read and agreed to the published version of the manuscript.

**Funding:** This research was funded by the National Nature Science Foundation of China (Grant No. 61631021).

**Conflicts of Interest:** The authors declare no conflict of interest.

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
