# Peer review of "Monolithic Integrated High Frequency GaN DC-DC Buck Converters with High Power Density Controlled by Current Mode Logic Level Signal"

_electronics, doi:10.3390/electronics9091540_

Round 1

Reviewer 1 Report

The presented idea is interesting and the results have great alignment in this paper. Here are my comments:

1. From table 1, it shows that although the highest in power density, but it gives up a lot of converter efficiency, especially between [10] and the presented one. What is the root cause of such a difference? Is it ready needed? Is seems this trace off is not worth having.

2.In Fig. 12, the point of [10] is missing.

3. A prototype photo is required.

4. Also, this paper is a lack of implementation detail. It is required for the reader, such as a parameter list, the detail specification, and the design strategy.

Reviewer 2 Report

Well written paper. It will be great if Fig. 2 is expanded for better visibility. Please add more x-axis ticks Fig. 4. The inset in Fig. 4 (c) is unreadable. Please update the title using the term 'current mode logic' instead of 'CML'. 

Round 2

Reviewer 1 Report

Thanks for the update. I have no more comments.